

# Oceanographic monitoring in Hornsund fjord, Svalbard

Meri Korhonen[1], Mateusz Moskalik[1], Oskar Głowacki[1], and Vineet Jain[1]

[1]Institute of Geophysics Polish Academy of Sciences, Księcia Janusza 64, 01-452 Warsaw, Poland

**Correspondence:** Meri Korhonen (mkorhonen@igf.edu.pl)

**Abstract.** Several climate-driven processes take place in the Arctic fjords. These include ice-ocean interactions, changes in biodiversity and ocean circulation patterns, as well as coastal erosion phenomena. Conducting long-term oceanographic monitoring in the Arctic fjords is, therefore, essential for better understanding and predicting global environmental shifts. Here we address this issue by introducing a new hydrographic dataset from Hornsund, a fjord located in south-western part of Svalbard archipelago. Hydrographic properties have been monitored with vertical temperature, salinity and depth profiles in several locations across the Hornsund fjord from 2015 to 2023. From 2016 onward dissolved oxygen and turbidity data are available for the majority of casts. The dataset contributes to the so far infrequent observations especially in spring and autumn and extends the observations typically concentrated in the central fjord to the areas adjacent to the tidewater glaciers. Because sediment discharge from glaciers and land is an inseparable part of the glacier-ocean interactions, the suspended sediment concentration in the water column as well as the daily sedimentation rate adjacent to the tidewater glaciers are monitored with regular water sampling and bottom-moored sediment traps. Here we present the planning and execution of the monitoring campaign from the collection of the data to the post-processing methods. All datasets are publicly available at the repositories referred to in the Data availability section of this manuscript.

## 1 Introduction

### 1.1 Hydrographic and climatic conditions in the West Spitsbergen fjords

Hornsund is the southernmost fjord in Spitsbergen, the largest island of Svalbard archipelago (Fig. 1a). The climate and hydrographic conditions in the fjord are strongly influenced by the topographically steered West Spitsbergen Current and the warm Atlantic Water it transports northward along the West Spitsbergen shelf break (Saloranta and Svendsen, 2001; Schauer et al., 2004). The heat loss from the West Spitsbergen Current to the atmosphere and surrounding water masses keeps the continental shelf west of Spitsbergen ice free. This results in warmer and more humid conditions compared to the north-eastern part of Svalbard, which is influenced by Arctic air mass and sea ice cover.

On the West Spitsbergen Shelf, the Sørkapp Current transports cold and fresh Arctic waters and sea ice around the southern cape of Spitsbergen (Fig. 1a). The Polar front between these waters of Arctic origin and the warm West Spitsbergen Current centered on the slope restricts direct intrusion of the pure Atlantic Water into the fjords (Saloranta and Svendsen, 2001). Large-scale atmospheric forcing and Ekman transport can, however, cause the Atlantic Water transported by the West Spitsbergen Current to flood onto the West Spitsbergen Shelf and mix with the Sørkapp Current (Nilsen et al., 2016; Goszczko et al., 2018).



The mixing with waters of Arctic origin results in modified or transformed Atlantic Water. It is this modified Atlantic Water that is most often observed inside the fjords and therefore generally responsible for potential heat transport towards the glacier termini (Svendsen et al., 2002; Nilsen et al., 2008; Promińska et al., 2017; De Rovere et al., 2022).

Calving rates at tidewater glaciers in the West Spitsbergen fjords have been identified to increase with increased sea water temperatures (Luckman et al., 2015; Holmes et al., 2019; van Pelt et al., 2019; Błaszczyk et al., 2023). As the Arctic continues to warm, accelerated frontal calving as well as enhanced surface and submarine melt from marine-terminating glaciers increase freshwater and sediment discharge into the fjords. Consequently, the vertical stratification, optical properties, nutrient availability and chemical composition of sea water are changing with potentially prominent consequences to marine ecosystems 35 (Murray et al., 2015; Meire et al., 2017; Moskalik et al., 2018; Błaszczyk et al., 2019; Hopwood et al., 2020).

## 1.2 Recent changes in the hydrography of West Spitsbergen fjords

Of the West Spitsbergen fjords, Hornsund is relatively little studied. More comprehensive observations and analyses are available from the more northern fjords, Isfjorden and Kongsfjorden (Fig. 1a). Studies from Isfjorden and Kongsfjorden indicate that the intrusion of Atlantic-derived waters, typically first observed in mid-June after the geostrophic control between the 40 fjord and shelf waters has weakened, reaches its maximum volume inside the fjord as late as September (Svendsen et al., 2002; Cottier et al., 2005; Nilsen et al., 2008; Pavlov et al., 2013). In recent years warming of both atmosphere and ocean has lead to reduced ice formation and decreased production of dense Winter Cooled Water (Muckenhuber et al., 2016; Tverberg et al., 2019). Consequently, the horizontal density gradient between the West Spitsbergen fjords and the shelf has weakened, allowing the warm shelf waters to penetrate into the fjords (Tverberg et al., 2019). Simultaneously atmospheric forcing during winter 45 has become more favourable for the flooding of Atlantic Water and less favourable for the presence of the Arctic Water on the West Spitsbergen Shelf (Nilsen et al., 2016; Goszczko et al., 2018; Strzelewicz et al., 2022). Together these changes have enabled the waters with Atlantic origin to occupy larger extent and to be more frequently observed in Isfjorden and Kongsfjorden especially during winter (Cottier et al., 2007; Nilsen et al., 2016; Tverberg et al., 2019; Skogseth et al., 2020; De Rovere et al., 2022). Strzelewicz et al. (2022) did not identify a trend in the presence of Atlantic waters in Hornsund, although they noted 50 a decrease in the presence of Winter Cooled Water. Their results were, however, based on a single transect acquired during summer. Overall, majority of the studies focusing on the hydrography of Hornsund are based on observations obtained in July. This leaves seasonal changes in hydrographic conditions undetected and therefore the full extent of Atlantic heat import may not be captured. In addition, analysis based on such temporally limited datasets causes a risk of interpreting temporal shifts in seasonal cycle as artifacts of interannual variability.

Another shortcoming of the regular observations available from Hornsund is that they are concentrated merely in the central fjord, ignoring the complex shoreline of Hornsund (Fig. 1c). Consequently, the seasonal and interannual variability of hydrographic conditions in the inner basins directly influenced by tidewater glaciers remain unexplored. The bathymetric sills at the entrance of the inner bays (Fig. 1c) potentially delay or restrict the influence of intruding Atlantic waters and hence the heat flux to the glacier termini (Nilsen et al., 2008; Arntsen et al., 2019). Arntsen et al. (2019) used mooring data at a sill of Brepollen, 60 the largest and innermost bay in Hornsund, to study the controls of the inflow of warm Atlantic water and its influence on the





tidewater glaciers. Their analysis covered two contrasting winters (1) 2010-2011, when waters at the sill were relatively cold, and (2) 2013-2014, when warmer waters were present. They concluded that the wind stress, coastal trapped waves and tides are the important mechanisms governing the cross-sill transport of watermasses from main basin to Brepollen. In addition, studying marine-terminating glaciers in western Spitsbergen, Holmes et al. (2019) showed that measuring temperature in the
waters directly in contact with, or as close as possible to, the glacier is essential for correctly estimating the impact of oceanic heat flux on frontal ablation.

### 1.3 General concept of the long-term oceanographic monitoring of Hornsund fjord

In order to address the need for more extensive seasonal and areal coverage of hydrographic observations, Institute of Geophysics Polish Academy of Sciences (IG PAS) initiated a campaign for the long-term oceanographic monitoring of Hornsund,
the so-called LONGHORN program. Hydrographic survey, consisting of vertical salinity and temperature profiles, has been carried out since 2015. From 2016 onward most of the profiles include measurements of dissolved oxygen and turbidity as well. The design of the survey takes into account the complex shoreline characteristic of Hornsund by covering areas close to the glacier termini as well as the central basin. In addition to the hydrographic observations, the monitoring includes regular measurements of suspended sediment concentration and sedimentation rate. The survey contributes to the availability of ob-
servations over spring, summer and autumn seasons. Here we present data collected until 2023. However, the campaign is still ongoing and its aim is to build a long-term dataset.

In Section 2 we present the study area, Hornsund fjord, and its distinct characteristics compared to the other West Spitsbergen fjords. In Section 3 we outline the monitoring campaign by reviewing the practical arrangements for field work including logistical challenges and the instruments used for measurements. We also give an overview on the temporal and spatial coverage
of the dataset. In Section 4 we describe the methods used for data-processing, including procedures for compression and quality control of the collected data. Finally, in Section 5, we present selected preliminary findings to highlight the benefits and importance of the dataset.

## 2 The study area

Hornsund is a shallow fjord with maximum depth of 240 m and length of 35 km. There is no bathymetric sill between the
approximately 8 km wide fjord mouth and the Western Spitsbergen Shelf, but on the other hand there is no deep connection either (Fig. 1c). Despite its southern location in the Svalbard archipelago, Hornsund is less influenced by the Atlantic waters than for example the more northern Isfjorden and Kongsfjorden (Nilsen et al., 2016; Promińska et al., 2017). One reason behind greater isolation of Hornsund is the much shallower, about 200 m deep, entrance compared to other West Spitsbergen fjords. Another factor contributing to the lesser presence of Atlantic waters in Hornsund is the more pronounced Polar front separating
the warm West Spitsbergen Current and the colder Sørkapp Current on the southern West Spitsbergen Shelf (Saloranta and Svendsen, 2001; Promińska et al., 2018).

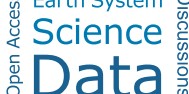

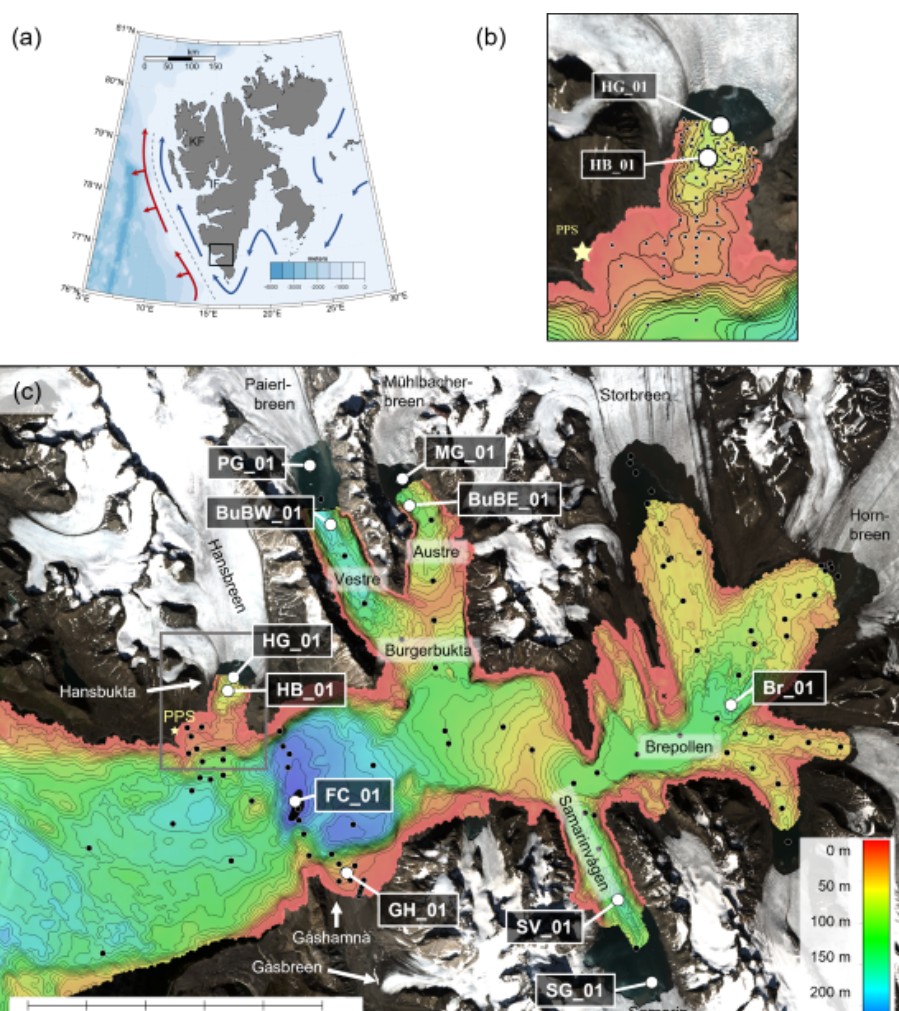

**Figure 1.** (a) Svalbard archipelago and the location of Hornsund fjord framed with black box. Locations of Kongsfjorden (KF) and Isfjorden (IF) are indicated. Schematic illustration of the major currents, the West Spitsbergen Current and the East Spitsbergen current, with red and blue arrows, respectively. The Polar front separating them is marked with a dashed gray line. Currents and front based on Saloranta and Svendsen (2001) and references therein. (b) Station plan for Hansbukta. The zoomed in area is indicated as a gray rectangle in c. The bathymetric scale is the same as in C. (c) Map of Hornsund area with locations of CTD stations included in the hydrographic survey. White dots indicate locations of the core stations since 2022. Black dots indicate the supplementary CTD stations. The yellow star denotes the location of the Polish Polar Station Hornsund. Bathymetric data is based on The Norwegian Hydrographic Service, Moskalik et al. (2013) and Ćwiąkała et al. (2018). Note that the bathymetric data close to the glaciers is missing. Landsat-9 image (taken on July 27, 2023) courtesy of the U.S. Geological Survey.





Hornsund has complex geometry with several inner bays, also called fjord arms, characterized by marine-terminating glaciers (Fig. 1c). Of the total surface area, approximately half consists of these smaller bays, which used to be almost entirely covered by glaciers in the beginning of the previous century (Błaszczyk et al., 2013). Błaszczyk et al. (2013) estimated that between 2001 and 2010 glaciers in Hornsund retreated with an average rate of 70 m per year while the average retreat for glaciers in Svalbard during the same period was 45 m per year. Still today Hornsund is heavily glaciated, which, together with the more restricted input from the West Spitsbergen Current, leads to higher freshwater flux and lower salinity compared to the other West Spitsbergen fjords (Promińska et al., 2017; Błaszczyk et al., 2019).

The five fjord arms - Hansbukta as well as Western and Eastern Burgerbukta in the north, Brepollen in the east and Samarin­vågen in the south (Fig. 1c) - are separated from the main basin by shallow sills originating from glacial deposit of moraine and therefore the warm and deep inflows into the main basin of Hornsund may have only limited influence on the glacier termini (Arntsen et al., 2019). It has been observed that the Winter Cooled Water typically persists in the deep depressions of the bays long into the summer and hence the bays can be regarded as an archive of conditions prevailing during the previous winter (Promińska et al., 2018; Skogseth et al., 2020).

The monthly mean air temperature in Hornsund varies from -10 °C in winter and early spring to almost 5 °C during sum­mer months (Wawrzyniak and Osuch, 2020). Since the 1970s the annual mean air temperatures in Hornsund have increased by 1.14 °C per decade (Wawrzyniak and Osuch, 2020). The most pronounced warming has occurred during winter with approxi­mately 4 °C increase between periods 1971-2000 and 2001-2015 (Isaksen et al., 2016). Observations of seawater temperature span considerably shorter time and show large interannual variability. Based on summertime observations, annual temperature increase of 0.03 °C was found for the period 2001-2015, but the trend was not statistically significant (Promińska et al., 2018).

## 3   The LONGHORN monitoring program

The oceanographic monitoring within the LONGHORN program is carried out by the Department of Polar and Marine Re­search at the Institute of Geophysics Polish Academy of Sciences (IG PAS) using the infrastructure of the Polish Polar Station Hornsund.

The monitoring consists of the following components:

- Vertical CTD (conductivity-temperature-depth) profiles

- Vertical profiles of dissolved oxygen and turbidity (available for the majority of CTD casts)

- Suspended sediment concentration with loss on ignition from water samples

- Daily sedimentation rate (sediment flux) with loss on ignition from bottom-moored sediment traps

Each component of the monitoring program is discussed individually in the following subsections.



## 3.1 Hydrographic monitoring

Initially the hydrographic monitoring program consisted of over 50 standard CTD-stations, which composed along and across fjord-sections in the main basin and in all the inner bays (Fig. 1 and Table 1). However, the data collected during the first three years indicated that having such a densely spaced measurement plan was not necessary to resolve the horizontal variability in Hornsund (Fig. 2). Therefore, from 2019 onward, the number of stations was reduced to 16. From 2022 onward the survey has consisted of 11 stations: one in the center of the fjord, one in Brepollen, one in Gåshamna and two stations in each of the other bays: Samarinvågen, Eastern and Western Burgerbukta as well as Hansbukta (Fig. 1c). Of the two stations in the major fjord arms, one is located in the center of the bay and the other one close to the glacier front.

The locations for the new stations since 2022 are chosen as the deepest points in the main fjord and the inner bays. Therefore their position somewhat differs, at most 1.5 km, compared to the stations from the initial phase of the monitoring program (Table 1). Nevertheless, we chose not to rename the stations as the horizontal variability even over larger distances was found to be negligible (Fig. 2). The stations close to the glacier front are located approximately 500 m from the edge of the glacier, equidistant from the shorelines. The distance to the glacier front is measured with a Bushnell Tour V5 laser rangefinder. Due to the retreating glaciers, the locations of these stations are not stationary, but change seasonally and interannually. Their positions shown in Fig. 1c are based on single, randomly chosen, measurements obtained during 2022.

The 11 stations measured from 2015 to 2023 are the core stations of the hydrographic monitoring program. The rest, about one hundred stations, constitute the supplementary stations which were visited mainly during the initial stage of the monitoring program and intermittent campaigns related to various research projects. Over one third of the supplementary stations are located in Hansbukta. The supplementary stations are shown in Fig. 1b and c as black dots.

Naming of the core stations follows the convention where the first letters indicate the region, for example BuBE for Eastern Burgerbukta and HB for Hansbukta. The stations 500 m from the glacier edge get the first letter from the name of the glacier, PG for example standing for Paierl glacier (Paierlbreen). The capital G stands for Glacier.

### 3.1.1 Description of the CTD-instruments

Different instruments, two SAIV A/S SD208 STD/CTDs and one Valeport miniCTD, are used for the CTD survey. All instruments measure directly pressure, in situ temperature and conductivity. In 2015 all observations were made with Valeport miniCTD, but since 2016 SAIV A/S SD208 CTD has been the most commonly used instrument as, in addition to temperature and conductivity, it includes sensors for turbidity and dissolved oxygen. In July 2018 the SAIV A/S SD208 CTD instrument was lost and measurements until the following June were carried out with Valeport miniCTD. A new SAIV A/S SD208 CTD has been the main device for hydrographic monitoring since June 2019.

SAIV A/S SD208 CTD is programmed to measure with frequency of 0.5 Hz. In 2016, the first year the sensor was used, higher sampling rate of 1 Hz was applied, but this compromised the auto-scale settings of the turbidity sensor and it was decided to decrease the sampling frequency. Valeport miniCTD is typically programmed to measure with frequency of 8 Hz. Other essential technical details of the CTD sensors are summarized in Table 2.



**Table 1.** List of core CTD-stations included in the hydrographic monitoring. In 2022 stations were relocated and the distance indicates difference netween the past and current positions. In addition new stations 500 m from the glacier termini were introduced. It should be noted that the positions of these stations are not fixed. The stations where regular monitoring of suspended sediment concentration and daily sedimentation rate are monitored are in bold.

| Station | Position 2015-2021 | Position since 2022 | Change in position/ Distance (m) | Bottom depth (m) | Number of casts |
|---|---|---|---|---|---|
| FC_01 | $76°58.73'N; 15°44.47'E$ | $76°58.44'N; 15°44.64'E$ | 530 | 242 | 56 |
| **HB_01** | $77°01.00'N; 15°38.17'E$ | $77°00.99'N; 15°37.95'E$ | 95 | 88 | 154 |
| HG_01* | - | $77°01.30'N; 15°38.51'E$* | - | 65-81 | 31 |
| BuBW_01 | $77°04.04'N; 15°50.03'E$ | $77°04.77'N; 15°48.64'E$ | 1460 | 175 | 40 |
| PG_01* | - | $77°06.13'N; 15°46.61'E$ | - | 87-163 | 11 |
| BuBE_01 | $77°04.84'N; 15°58.97'E$ | $77°05.18'N; 15°56.75'E$ | 1120 | 123 | 40 |
| MG_01* | - | $77°05.79'N; 15°56.05'E$ | - | 71-105 | 15 |
| Br_01 | $77°00.15'N; 16°27.69'E$ | $77°04.46'N; 16°29.15'E$ | 830 | 148 | 51 |
| SV_01 | $76°56.12'N; 16°17.12'E$ | $76°56.06'N; 16°17.23'E$ | 130 | 159 | 38 |
| SG_01* | - | $76°54.15'N; 16°20.44'E$ | - | 25-90 | 14 |
| **GH_01**[1] | $76°56.82'N; 15°49.35'E$ | $76°56.79'N; 15°49.79'E$ | 200 | 30 | 35 |

\* These stations are located 500 m from the glacier termini and therefore the position is changing with glacier retreat/advance. The coordinates are chosen randomly from data collected in 2022. The bottom depth is given as minimum and maximum depths.

[1] Station GH_01 was introduced to the monitoring program in 2021.

All instruments used to measure hydrographic properties were laboratory calibrated by their respective manufacturers before
purchase. After this initial calibration, the instruments are intermittently compared against each other. An extensive intercomparison of all instruments was conducted in May 2018. The results of the intercomparison are discussed in the supplementary material.

### 3.1.2   Collection of hydrographic data

The hydrographic monitoring is conducted by trained research personnel of the Polish Polar Station Hornsund from an outboard
motorboat. To obtain a full-depth profile the instrument is typically lowered all the way to the bottom. It should be noted that until summer 2016, due to the length of rope used, the maximum depth of casts was only 150 m and therefore the bottom was not reached in some locations. The lowering speed is not intended to exceed $0.5\,\mathrm{m\,s^{-1}}$. However, usually the rate of descent is manually controlled and only occasionally a winch with rope speed monitor has been available. Manual operation causes irregularities in the descent rate which are transferred to the sampling frequency per depth unit. The median lowering
speed calculated from the raw data is $0.4\,\mathrm{ms^{-1}}$ for SAIV A/S SD208. The highest median descent rate for a single profile was $0.9\,\mathrm{ms^{-1}}$ and the highest instantaneous lowering speed reached $1.8\,\mathrm{ms^{-1}}$.

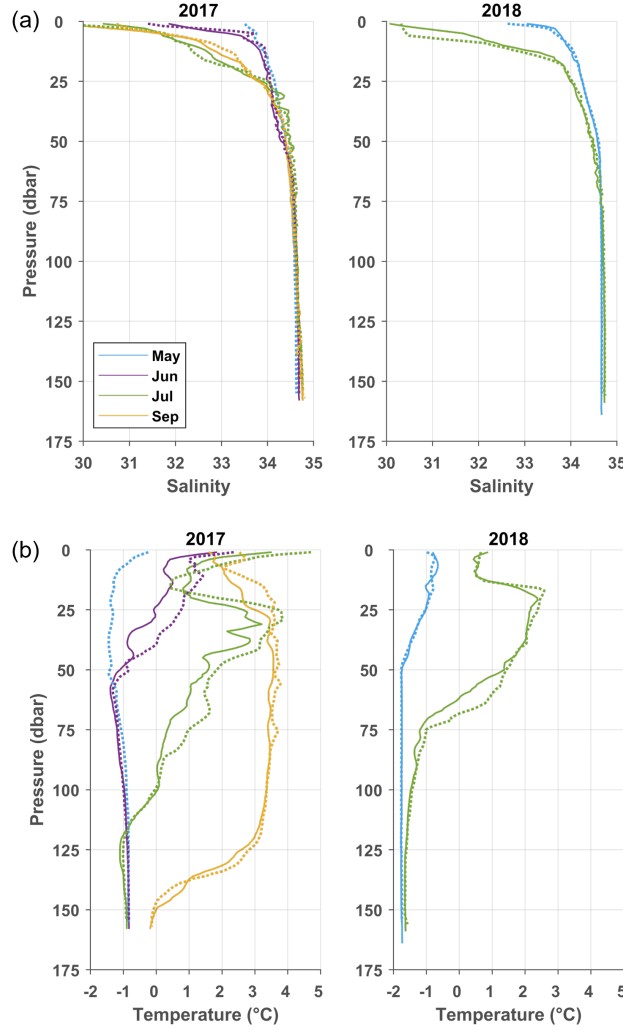

**Figure 2.** Salinity (a) and temperature (b) profiles displaying the modest horizontal variability of hydrographic properties within Vestre Burgerbukta. Stations BuBW_01 (solid line) and BuBW1_04 (dotted line) are shown for Vestre Burgerbukta. Distance between the two stations is 2200 m with BuBW_01 being closer to the glacier front. Two different years (2017 and 2018) with observations covering several months are shown.

Other potential source of inaccuracy caused by the measurement arrangements is the drift of the boat from the measurement point during the cast. In general, when considerable (over 100 m) drift from the fixed coordinate point is observed to occur, the cast is repeated after repositioning. The same 100 m distance is considered as the maximum acceptable deviation from the fixed location set for the monitoring station. Therefore the maximum error to the given position is considered to be less than 100 m. However, in addition to the drift of the boat, due to the light weight of the instrument, currents may cause tilt in the rope.





**Table 2.** Instruments used in hydrographic monitoring and the directly measured variables (information obtained from the manufacturers).

| Instrument | Measured property | Unit | Accuracy |
|---|---|---|---|
| SAIV A/S SD 208 | Pressure | dbar | 0.01 % of the full-scale range |
| | Temperature | $^\circ$C | 0.002 $^\circ$C |
| | Conductivity | $\mathrm{mS\,cm^{-1}}$ | 0.003 $\mathrm{mS\,cm^{-1}}$ |
| | Dissolved oxygen | $\mathrm{mg\,l^{-1}}$ | 0.5 $\mathrm{mg\,l^{-1}}$ |
| | Turbidity | FTU | linearity <2 % |
| Valeport miniCTD | Pressure | dbar | 0.05 % of the full-scale range |
| | Temperature | $^\circ$C | 0.01 $^\circ$C |
| | Conductivity | $\mathrm{mS\,cm^{-1}}$ | 0.01 $\mathrm{mS\,cm^{-1}}$ |

Overall, the inaccuracies resulting from positioning or tilting are considered to be negligible because of the shallow depths and rather small horizontal differences in the water column structure in different parts of Hornsund fjord (Fig. 2).

### 3.1.3 Horizontal and temporal coverage of the dataset

Because of the polar night, weather and ice conditions, the timing of boat-based observations is limited between May and October, although occasional observations during the polar night are also available. The availability of hydrographic observations from different areas of Hornsund fjord are shown in Fig. 3. The most comprehensive and frequent hydrographic observations have been obtained in Hansbukta due to its proximity to the Polish Polar Station Hornsund. Of the larger fjord arms, the most extensive dataset is obtained from Brepollen.

The desired interval for hydrographic measurements varies from two weeks for the stations close to the Polish Polar Station (Hansbukta, Central fjord and Gåshamna) to four weeks for the other stations. However, the frequency with which each station is visited depends on weather and ice conditions, availability of staff and the subjective commitment of the person responsible for the observations. Consequently, the timing of observations varies between the years and monthly observations are not available for each year and area.

### 3.2 Suspended sediment concentration and sedimentation rate

With increasing freshwater discharge from glaciers and land, the coastal waters are receiving more terrestrial matter (Zhang et al., 2022). Therefore the LONGHORN monitoring campaign includes two core stations in contrasting environments for regular sediment sampling. Hansbukta (station HB_01) receives freshwater and sediments from the marine-terminating Hansbreen, whereas Gåshamna (station GH_01) is a shallow bay receiving river transport from Gåsbreen, the largest land-based glacier in

Hornsund, and the surrounding terrain (Fig. 1c and Table 1). The aim is to obtain observations covering the period from May to October, but there is some variability in the start and end timing of the monitoring each year. In addition to the stations in Hansbukta and Gåshamna, occasional samples are available from the other fjord arms as well (Fig. 4).

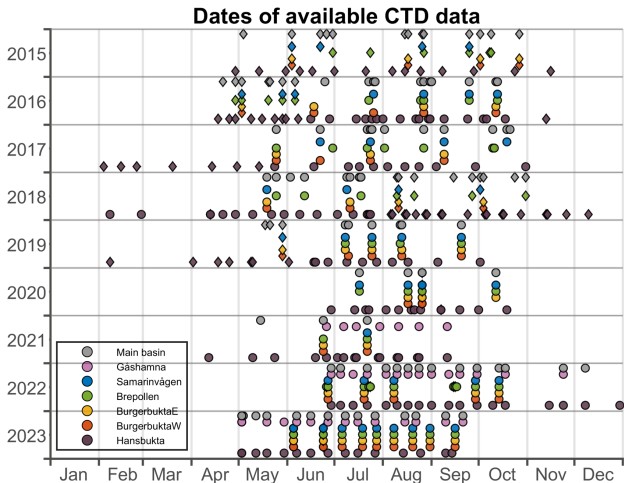

**Figure 3.** Temporal coverage of CTD data. Dates of CTD casts obtained with Valeport miniCTD are marked with diamonds and casts obtained with SAIV A/S SD208 CTD, equipped with turbidity and oxygen sensors, are shown with circles.

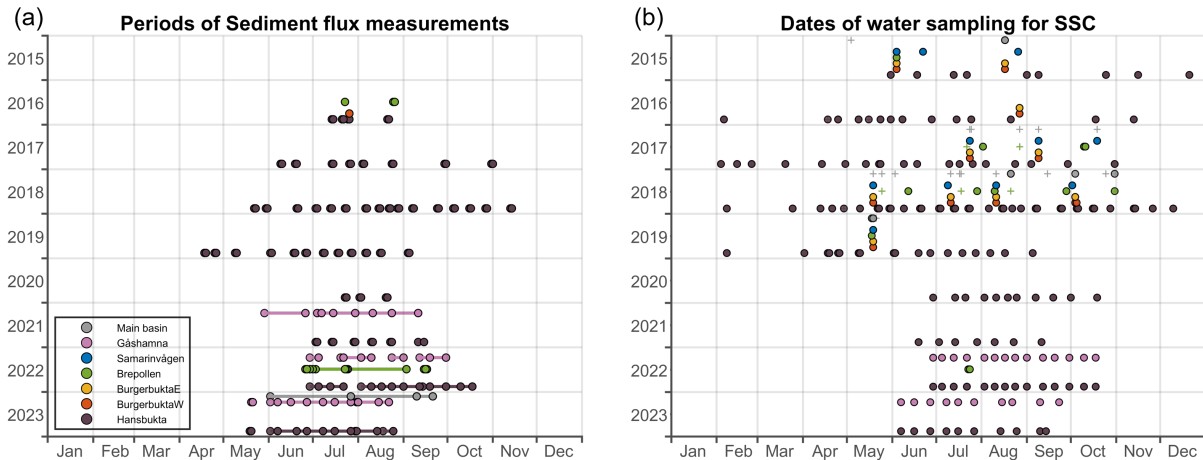

**Figure 4.** (a) Periods of sediment flux measurements. The circles connected with a line denote the time between sediment trap deployment and recovery. When no line is visible the sediment trap was deployed for a short time (in general for 24 h). (b) Dates of water sampling for suspended sediment concentration. The circles denote dates when multiple depths in the water column were sampled for analysis of suspended sediment concentration. The plus-signs mark the dates when only a surface sample (0 m) was obtained.

Regular water sampling and sediment flux measurements at the station HB_01 in Hansbukta begun in 2016. In 2015 measurements were conducted slightly further from the glacier. For the early years of monitoring (2015-2018) suspended sediment concentrations are available from the section perpendicular and also from section parallel to the glacier front (Fig. 1b). Monitoring in Gåshamna started in 2021 with measurements on sediment deposition. Since 2022 both sedimentation rate and



suspended sediment concentration are available from Gåshamna. It should be noted that the location of the station GH_01 differs by 200 m between 2021 and 2022.

The water samples are collected with a Hydrobios Free Flow 1 l Niskin Bottle from multiple depths in the water column. The daily sedimentation rate is measured with a sediment trap consisting of two cylindrical tubes with diameter of 80 mm and volume of 1 l. The tubes are mounted on a gimballed frame moored to the sea floor. In order to keep the tubes vertically oriented, a submersible float is attached 5 m above the trap. During the first three years (2016-2018) of sediment flux measurements the sediment traps were in general deployed above and below the pycnocline, 5-15 m and 20-25 m from the surface. Since 2019 most of the measurements are obtained 5 m above the bottom. The material settled to the sediment traps was initially collected after 24 h, but since 2022 the sediment traps are deployed for approximately 10 days.

Typically the whole 1 l water sample is filtered through Whatman GF/F 0.7 filters (diameter 45 mm) for the analysis of suspended sediment concentration. For samples from the sediment traps, partitioning of the 1 l sample is made before filtration. Typically 1/4, 1/8 or 1/16 of the sample is filtered. Prior to filtration the filters are dried in a $200\,^\circ$C oven for 2 hours and weighed on a $1.00 \times 10^{-3}$ g precision scale. After filtration the filters are dried at $40\,^\circ$C for 24 hours and left in the desiccator before weighing. Sediment weight is obtained from the difference in the dry filter weight before and after filtration. To get the suspended sediment concentration, this mass is divided by the volume of the water sample used for filtration. The daily sediment deposition is calculated from the obtained sediment weight divided by the area of the tube and the duration of deployment.

As a final step organic matter is incinerated by baking the filters in $550\,^\circ$C oven for 3 hours and left in the desiccator before weighing. The loss on ignition (LOI) value is calculated as the difference in the weight of sediments before and after ignition divided by its weight before ignition. This value can be considered as an estimate of the amount of organic content in the sediment. The final dataset on sedimentation rate is available at https://doi.pangaea.de/10.1594/PANGAEA.967172 (Moskalik et al., 2024c). It includes information on location, deployment and recovery dates, deployment depth, sedimentation rate and the LOI value. The dataset on suspended sediment concentration is found at https://doi.pangaea.de/10.1594/PANGAEA.967173 (Moskalik et al., 2024d) with information on location, sampling date and depths as well as concentration of total suspended particles and the LOI value.

## 4 Processing of CTD data

### 4.1 Removal of unrealistic values

The data recorded by SAIV A/S SD208 CTD and Valeport miniCTD are downloaded and converted into ASCII-format with Minisoft SD200W and DataLog x2 softwares, respectively. Both softwares automatically apply their own routines to calculate thermodynamic variables (potential temperature, practical salinity, density and sound velocity) based on the directly measured variables (conductivity, in situ temperature and pressure). However, for consistency, all derived variables in the final processed dataset are calculated using Gibbs Seawater (GSW) Oceanographic toolbox developed for Matlab (McDougall and Barker, 2011). The GSW Toolbox implements the International Thermodynamic Equation of Seawater-2010 standard (TEOS-10, IOC,





SCOR and IAPSO (2010)). It should be noted that the variables in the raw data files are those derived by the softwares provided by the manufacturers of the instruments.

Automatic routines are applied to filter out salinity values lower than 20 and higher than 35.2. High salinity spikes occur rarely and almost all filtered values are unrealistically low and occur close to the surface or in the bottom layer when the sensor may be in contact with the bottom sediments. The suspiciously low salinity values are also associated with atypically large

vertical gradients ($>1\,\mathrm{PSU\,dbar^{-1}}$) and mostly occur in early spring when there is practically no freshwater flux to the fjord. In the bottom layer the low salinites are identified also by reversed density stratification. In such cases all bottom values are discarded.

The measured temperature is tested against the local freezing point temperature and when the temperature is more than 0.1 degrees below the freezing point, both temperature and salinity are removed. This typically occurs in the topmost surface layer

in early spring when the air temperature is below zero. Temperature below freezing point is often associated with suspiciously low salinity, which supports our conclusion that the salinities below 20 result from ice formation in the conductive cell.

Occasionally turbidity and oxygen sensors show spikes close to zero likely due to malfunction in the connective cables. These spikes are removed by setting a lower limit of 0.2 for turbidity and 65% for dissolved oxygen saturation. Typically turbidity does not exceed 150 FTU and therefore all profiles showing higher values are visually inspected. Upper acceptable

limit for oxygen saturation is set to 140 %. The upper limit is exceeded when the protective cap of the oxygen sensor is not removed before the cast. In addition to the high oxygen concentration, these casts can be visually recognized from rapidly increasing oxygen saturation with depth.

Turbidity and oxygen sensors are located in the base of the instrument and therefore they may become submerged in the bottom sediments when the instrument is lowered. Because these sensors are optical, the high sediment concentration results

in rapid beam attenuation and near-bottom peaks especially in the downward looking oxygen sensor. Such peaks are detected and removed.

## 4.2 Data compression

After deriving the thermodynamic variables as well as initial filtering of unrealistic values, the data is compressed to 1 dbar vertical bins. This is done by taking the median value of measurements within half-unit distance from the center of the bin.

Consequently, because the first bin is centered at the depth of 1 dbar, the measurements in the uppermost 0.5 dbar layer are not included in the final averaged data, but can be found in the raw data files. In general both down and up casts are used in the median-filtering for temperature and salinity and the thermodynamic variables. However, all profiles are visually checked and occasionally the up-profile is rejected because of values deviating from those of the down-profile. This often occurs with data recorded by Valeport miniCTD as the conductivity sensor is located in the base of the instrument and can therefore become

clogged when in contact with bottom sediments. Of the salinity profiles obtained by SAIV A/S SD208 CTD less than 1 % of the up-casts are rejected.

For turbidity and oxygen typically only the down profile is included in the final median-filtered dataset, because possible bottom contact often leaves the optical sensors covered in sediments for a considerable part of the ascend. Despite this, occa-





sionally both, down- and up-casts, or only the up-cast have been used for turbidity and oxygen. This may be the case when the
instrument has been lowered too fast and the overall quality of the data could be improved by including the up-cast. In addition,
in a few cases the sensors may not have been adequately cleaned before the cast, which leaves in particular the turbidity sensor
to display suspiciously high values during the down-cast. In such cases only the up-cast has been considered. Nonetheless,
before accepting the up-profile, the data is always carefully visually inspected.

### 4.3   Final steps: smoothing and data archival

Considering the programmed sampling rate and estimated lowering speed of $0.4\,\mathrm{m\,s^{-1}}$, the median is typically taken from two
or three measurements when only downcast is included. Due to the relatively low number of measurements within one vertical
bin, averaging based on median-filtering does not necessarily remove individual spikes and outliers. Therefore, after linearly
interpolating over missing values, the data is smoothed using Matlab's *smooth*-function with lowpass filter and window size of
four (moving average of four consequent depths). If less than 70 % of the data is available over the total depth, interpolation
is not performed and the whole profile is rejected. An example of raw data and the final median-filtered and smoothed data is
shown in Fig. 5.

    Before making the data publicly available at a data repository, the data are visually inspected for consistency by comparing
group of profiles selected according to area or timing of observation. The CTD data is divided into two datasets: one consisting
of the core CTD stations and the other one including all other supplementary stations. Both processed final datasets include
variables shown in Table 3. Conductivity is not included in the final 1-dbar averaged data sets but can be found in the raw data
files. The core CTD dataset is available at https://doi.pangaea.de/10.1594/PANGAEA.967169 (Moskalik et al., 2024a) and the
supplementary dataset at https://doi.pangaea.de/10.1594/PANGAEA.967171 (Moskalik et al., 2024b).

## 5   Oceanographic data

### 5.1   Overview of hydrographic conditions in Hornsund

The hydrographic characteristics in the central Hornsund fjord in July have been extensively studied and described by Promińska
et al. (2017, 2018) and Strzelewicz et al. (2022). The dataset introduced here adds to the current knowledge by extending the
observations from the central fjord to the various fjord arms as well as by allowing assessment of seasonal variability. One
of the questions arising from the complex geometry of Hornsund is the mixing of the fresh and cold waters of glacial and
terrestrial origin with the saltier and warmer waters from the West Spitsbergen Shelf.

The effect of freshwater flux is seen in Fig. 6a showing the vertical salinity profiles obtained in August: In the bays close
to the glaciers the surface salinity is typically around 2 PSU lower than in the central fjord, where the surface salinity is in a
typical year close to 31. Whereas the surface salinity remains relatively stable between the years in the center of the fjord and
in Brepollen, in the western and eastern Burgerbukta local processes seem to cause considerable interannual variability.


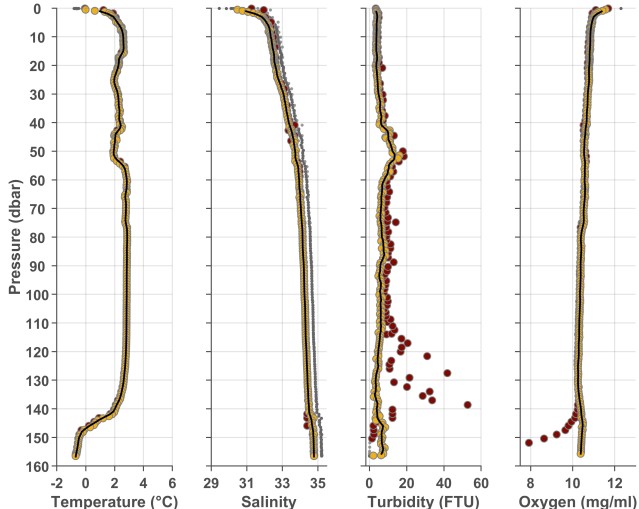

**Figure 5.** An example of processing the CTD-data from station BuBW_01 on September 20th, 2019. The raw data is marked with gray dots. The yellow and brown circles indicate the measurements taken during the down- and up-profiles, respectively. The black line is the final data after averaging to 1-dbar vertical bins and smoothing. Note that for the profile shown here the up-profile is rejected for turbidity and oxygen due to contact with bottom-sediments.

The fjord arms with a bathymetric sill are known to be archives of Winter Cooled Water. While the temperature in the central

fjord is over $2\,^\circ$C also in the deep layers, the bottom temperatures in all the larger fjord arms generally remain close to zero even in August (Fig. 6b). In the upper layers the temperatures in the bays typically reach 4 or $5\,^\circ$C by August. The surface temperatures observed in the inner bays are comparable to the temperature in the center of the fjord.

Already the timeseries from 2015 to 2023 shows high interannual variability of hydrographic conditions in Hornsund. For example, in August 2015 the upper 50 m layer was cold and fresh in all regions, but in 2016 the whole water column was

warmer and saltier than the average for the nine year period (Fig. 6). In August 2016 the warm signal was observed in the bottom layer in all basins with temperatures over $1\,^\circ$C higher compared to the other years. The bottom waters in the western Burgerbukta reached even higher temperature, exceeding $3\,^\circ$C.

Compared to the larger fjord arms, the smaller and shallower Hansbukta is more directly influenced by the water exchange with the central fjord. The highest salinity in the upper layers is observed in 2016 together with the high temperatures through-

out the whole water column. Despite the heat flux, likely of Atlantic origin, Hansbukta is clearly fresher than the other bays. The salinity of the water column at depths 25-75 m varies between 32 and 34 PSU, being about 1 PSU lower than the water at comparable depths in the other bays. Together with the high temperature, the low salinity may indicate higher freshwater flux from glacier melt compared to the other basins.



**Table 3.** Variables in the final processed dataset for vertical CTD-profiles. Additional metadata such as the used instrument and bottom depth can be found in the accompanied station list.

| Variable | Unit/format |
|---|---|
| Event label | - |
| Date and time [1] | ISO-8601 |
| Latitude | Decimal degree |
| Longitude | Decimal degree |
| Pressure, water | dbar |
| Depth, water | m |
| Temperature (in situ) | $^\circ$C, ITS-90 |
| Potential temperature | $^\circ$C |
| Practical salinity | unitless, PSS-78 |
| Density, $\sigma_\theta$ (0) | $\mathrm{kg\,m^{-3}}$ |
| Sound velocity | $\mathrm{m\,s^{-1}}$ |
| Turbidity [2] | FTU |
| Dissolved oxygen saturation [2] | % |
| Dissolved oxygen concentration [2] | $\mathrm{mg\,l^{-1}}$ |

[1] Start time of the profile. Valeport miniCTD data only include date.

[2] These variables are available for casts recorded by SAIV A/S SD208.

### 5.1.1 Seasonal variability of the hydrographic properties in Hansbukta

Seasonal variability in the hydrographic conditions is demonstrated with data collected in Hansbukta in 2017 (Fig. 7). In 2017 the water column was at the coldest, near the freezing point temperature, from late March until early May. First surface warming accompanied by freshwater input is observed during May. In June the oxygen concentration is still high ($14\,\mathrm{mgl^{-1}}$) with saturation exceeding 100 % in the upper 40 m. By the end of June, warming and freshening intensify and simultaneously the oxygen concentration starts to degrade.

The water column is at the warmest in August when the temperature reaches around 4 degrees throughout the water column. The surface temperatures remain slightly lower due to the input of fresh and cold water from glacier melt. It should be noted that Hansbukta is a rather shallow bay, with maximum depth less than 90 m, and therefore the mixing often extends all the way to the bottom. As a result there is no thermocline present during the warmest months, whereas the deeper bays are typically vertically well stratified (Fig. 6).

The transition back towards winter conditions begins in September when the surface temperature starts to decrease (Fig. 7a). However, salinity continues to decrease and the lowest salinity throughout the whole water column is observed in late September when the salinity varies from 31 at the surface to 32.5 in the deeper layers (Fig. 7b). The transport of glacial meltwater into


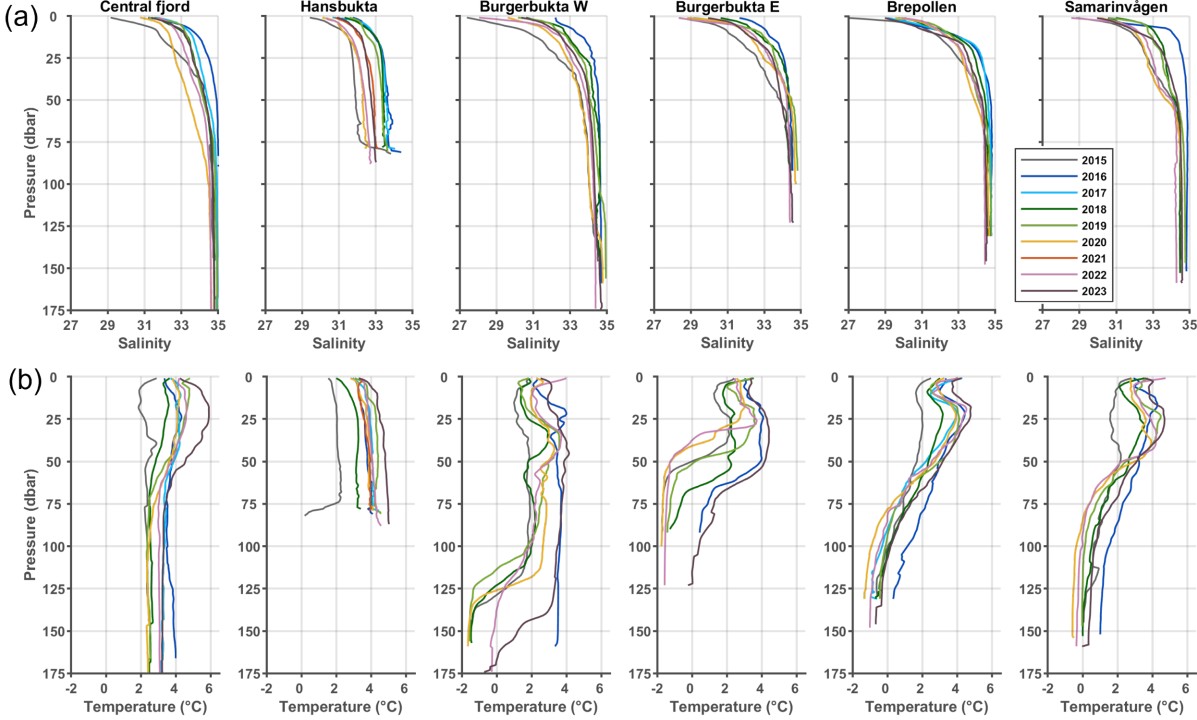

**Figure 6.** Interannual variability of salinity (a) and temperature (b) in different basins in Hornsund fjord from 2015 to 2023. Average profiles for August each year are shown. Note that the data availability varies between basins, which results in missing data during some years.

the fjord is reflected also in the distribution of suspended sediment concentration, represented as turbidity in Fig. 7c. Although the suspended sediment concentration peaks already in July near the surface, in September the sediment discharge is distributed throughout the water column.

## 5.2 Suspended sediment concentration (SSC) and sedimentation rate

### 5.2.1 Comparison of turbidity and suspended sediment concentration

Because collection and filtration of water samples is time consuming, turbidity sensors are used to obtain full depth vertical profiles of the sediment load in the water column. Turbidity describes the optical clarity of water, measured as FTU/FNU (Formazine Turbidity Unit or Formazine Nephelometric Unit). One FTU is defined as a response to $1 \, \text{mg} \, \text{l}^{-1}$ solution of Formazin. However, turbidity depends on the size and other light scattering properties of suspended particles. In Fig. 8 vertical turbidity profiles together with suspended sediment concentration from water samples are shown.

Most turbidity readings are below 20 FTU (Fig. 8). The typical maximum values of around 40-60 FTU and $\text{mg} \, \text{l}^{-1}$ are observed in the upper 10 m layer. It is in this upper layer where the readings from the turbidity sensor and the SSC from the water samples have the highest correspondence. Deeper in the water column the turbidity sensor generally gives lower values

Earth System
Science
Data

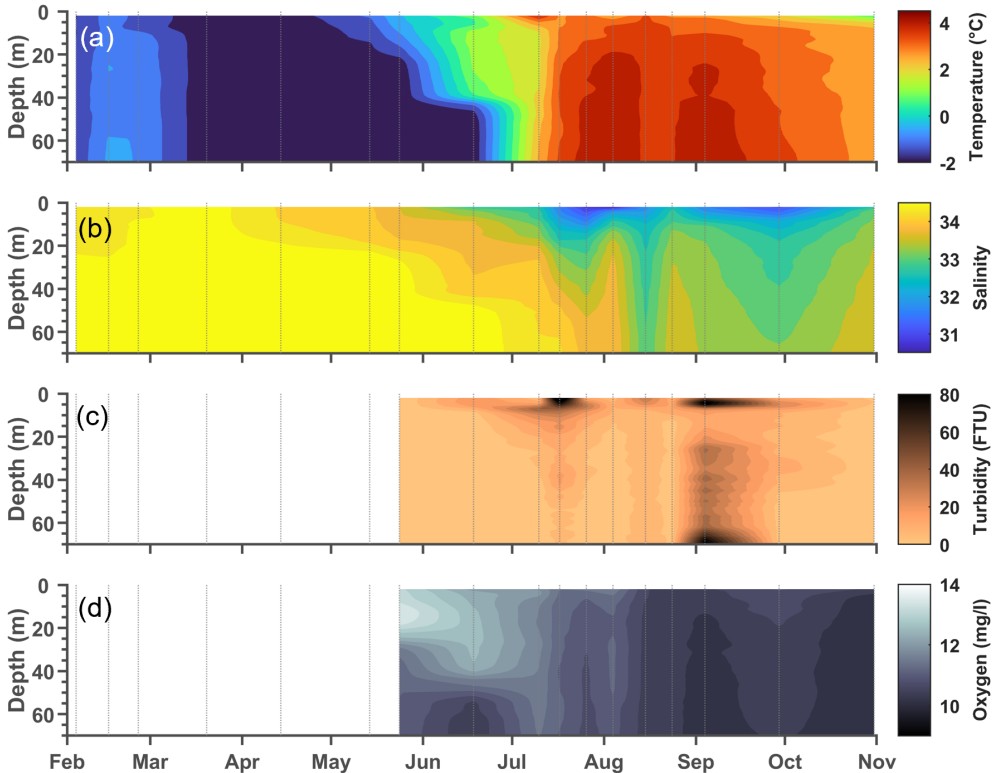

**Figure 7.** Seasonal variability of temperature (a), salinity (b), turbidity (c) and dissolved oxygen concentration (d) at the station HB_01 in Hansbukta during 2017. Turbidity and dissolved oxygen data are not available for the beginning of the year 2017. The thin vertical lines indicate dates when the vertical profiles were obtained.

compared to the water samples. For example, at depths 20, 40 and 60 m the median SSC obtained from the water samples suggests clearly higher sediment concentration than the turbidity sensor (Fig. 8).

We assume the difference between the turbidity sensor and the SSC from water sampling may be partly due to the autorange setting used during measurements. This means that the turbidity sensor detects the appropriate maximum turbidity and adjust 340 the accuracy according to that. Because of the generally low turbidity, the sensor typically uses the lowest available maximum, 12.5 FTU. It is thus possible that when turbidity exceeds this limit, the sensor readjusts the scale with delay. This may result in the sensor passing a rather thin layer of high sediment concentration recording it with inappropriate range.

An additional uncertainty is related to the sampling depths. Because the lowered Niskin bottle does not have a remotely readable pressure sensor, the sampling depth is estimated by the length of the rope used for collecting the sample. In deeper 345 layers, the currents or drift of the boat can result in considerable tilt of the rope. Consequently the sampling depth can be significantly shallower than indicated by the rope length.

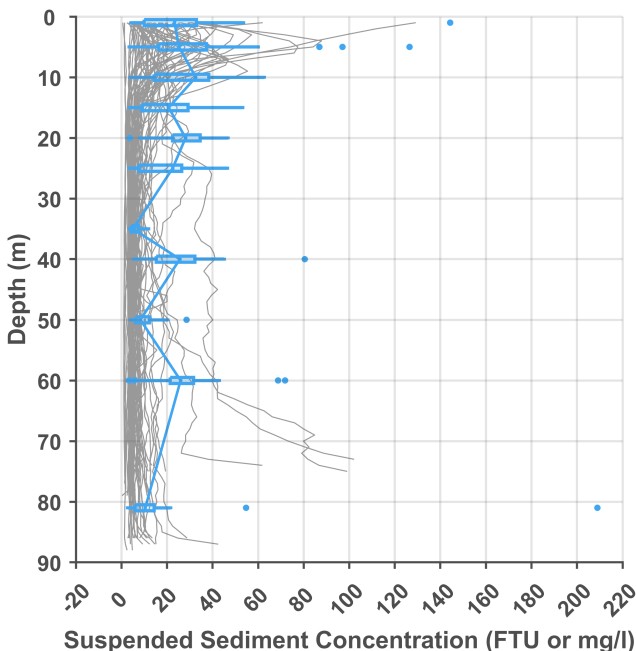

**Figure 8.** Comparison of turbidity profiles (gray) to suspended sediment concentration from water samples (blue). All vertical turbidity profiles obtained simultaneously with water sampling from Hansbukta are shown. For each depth of the suspended sediment concentration median value (blue vertical profile), interquartile ranges (blue horizontal boxes), minimum and maximum values (blue horizontal lines) as well as outliers (blue dots) are shown. The surface samples (depth = 0 m) are excluded. The depth of the samples obtained 5 m above bottom ranged between 78 and 83 m, but here they are all treated together as samples from depth 81 m.

### 5.2.2 Interannual and seasonal variability in suspended sediment concentration and sediment flux in Hansbukta

Suspended sediment concentration (SSC) above pycnocline (1-25 m) typically varies between 20 and 40 mg $l^{-1}$ from May to October (Fig. 9). On average about 20-30 % of the total suspended solids is organic. Although the SSC in close proximity of
tidewater glaciers is predicted to increase with accelerating glacier retreat (Zhang et al., 2022), the SSC in Hansbukta is clearly lower (10-25 mg $l^{-1}$) during the last three years (2021-2023) than during the preceding years 2015-2020.

Despite the interannual variability, seasonality can be detected from the timeseries (Fig. 9). The range of SSC within the pycnocline increases in July and August when also the concentration maxima are observed. The total mass of organic matter does not show clear seasonal signal. However, the proportion of organic matter is higher during spring and early summer.
During the last three years when the total SSC is low, also the mass of organic matter decreases albeit not as strongly as the amount of SSC.

The sediment flux maxima occur in July-August, simultaneously with the SSC maximum in the pycnocline. The maximum sediment fluxes amount to over 3000 g m$^{-2}$ day$^{-1}$, occasionally reaching nearly 5000 g m$^{-2}$ per day. For the rest of the season covered by observations the fluxes generally remain around 500-2000 g m$^{-2}$ day$^{-1}$. On average 5 % of the particulate matter

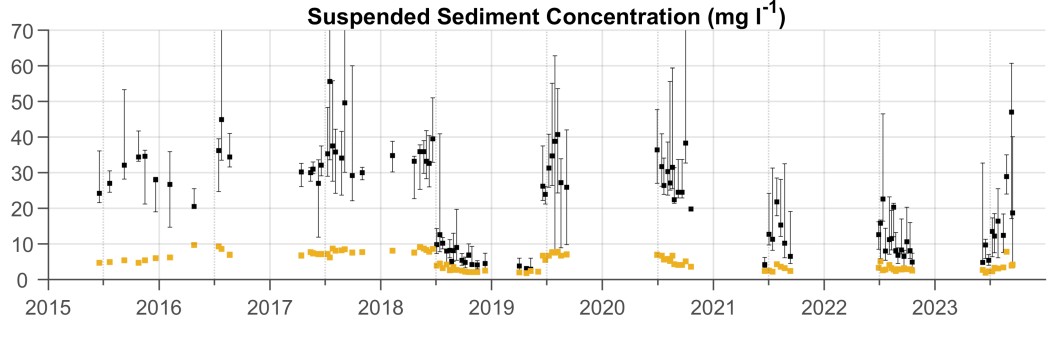

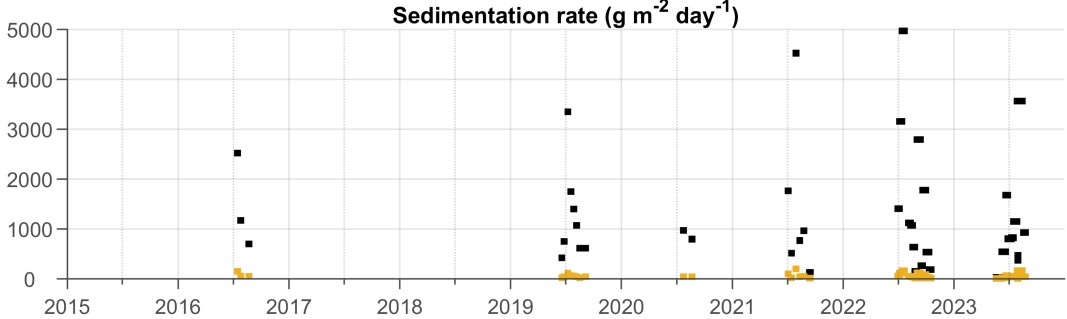

**Figure 9.** Upper panel: Suspended sediment concentration above pycnocline (1-25 m) at the station HB_01 in Hansbukta. The black squares mark the median SSC and the errorbars represent the minimum and maximum values. Note that some of the outliers are located outside of the figure axes. Lower panel: Average daily sediment flux 5 m above bottom in station HB_01 in Hansbukta. The yellow squares show the median amount of organic matter (LOI).

deposited at the bottom is organic. This is considerably less than the amount of organic matter above the pycnocline, indicating that most of the organic matter does not reach the bottom of the fjord at this location.

## 6 Conclusions and outlook

We present here an oceanographic dataset from Hornsund, a fjord in West Spitsbergen. The dataset consists of vertical salinity, temperature, turbidity and dissolved oxygen profiles across the fjord as well as sediment weights from water samples both from
different depths in the water column and from the sediment traps. The data is collected as a part of LONGHORN monitoring program that started in 2015 and is still ongoing. This new dataset improves availability of seasonal observations in the high-latitude fjords by providing information on the oceanic conditions prior to the summer warming, during the melt season and in late autumn. Observations spanning over several months benefit, for example, tracking of the frequency, timing and volume of the Atlantic water intrusions from the West Spitsbergen shelf. Monitoring the propagation of the seasonal signal is essential
also because the inner basins with sub-marine sills potentially delay the mixing of warm oceanic inflow with the colder waters adjacent to the glaciers.



Compared to other fjords in the western Spitsbergen, Hornsund is the southernmost but yet more restricted from the influence of the warm West Spitsbergen Current. Its unique geometry and multiple, fast retreating tidewater glaciers provide an ideal location for the studies at the interface between land, glaciers and ocean. The preliminary results from the first nine years of the
monitoring program show high interannual variability as well as considerable horizontal differences between the central basin and the inner basins. The data collected so far demonstrates the importance of capturing horizontal and seasonal variability in hydrographic conditions in order to better understand interactions between glaciers and ocean.

## 7    Code and data availability

Both, raw and processed, CTD-data as well as the suspended particulate matter and loss on ignition (LOI) values from filtration
of water samples are available from the data repository of IG PAS https://dataportal.igf.edu.pl/ as well as PANGAEA database:

- core CTD data, Moskalik et al. 2024a, temporary access key without logging in
  https://www.pangaea.de/tok/ff2d7cae4caf74f1d0ced10abdb74b86ea553883

- supplementary CTD data, Moskalik et al. 2024b, temporary access key without logging in
  https://www.pangaea.de/tok/d435cda9d594c9650df01f222977125a500b352f

- sediment flux, Moskalik et al. 2024c, temporary access key without logging in
  https://www.pangaea.de/tok/1c4eb8cbfa59320a1db28df347e25c49bd94a649

- suspended sediment concentration, Moskalik et al. 2024d, temporary access key without logging in
  https://www.pangaea.de/tok/418c50529e1283e442bdf142db901a92cc865eb3

The dataset will be updated annually to the IG PAS data repository.
Code to process the data and produce the figures was written with Matlab 2021, except for the maps in figures 1 b and c. The codes are available upon request from the corresponding author.

## Appendix A:  Correction of SAIV A/S SD208 conductivity sensor

Because of the detected offset in the conductivity sensor in SAIV A/S SD208 CTD instrument, a correction is applied to the readings before the data processing steps described above. Here we explain how the intercalibration experiment for quantifying
the offset was realized.

Our suspicion of potential offset in the conductivity sensor of SAIV A/S SD208 CTD was first raised by the differing results when compared to the salinity observations presented in Promińska et al. (2017). Therefore in May 2018 a 24-hour intercomparison with all available laboratory-calibrated sensors was realized in Isbjörnhamna, close to the Polish Polar Station. All instruments (including SAIV A/S SD208 SN 1269, Valeport miniCTD, three separate RBR CTDs and a See-Bird Electronics
SBE37SIP CT) were anchored at around 5 m depth, with maximum distance between instruments being less than 1 m. This



experiment revealed that readings from SAIV A/S SD208 conductivity sensor were indeed considerably higher than those from other sensors (not shown).

As the other conductivity sensors, apart from SAIV A/S SD208 (SN 1269), used in the 24-hour intercomparison displayed practically no differences, Valeport miniCTD was chosen as an accurate reference. In order to quantify the correction with varying temperature, salinity and pressure ranges, parallel vertical profiles with the two instruments were obtained on September 22nd, 2017 and April 13th, 2018 with SAIV A/S SD 208 (SN 1269). Because this instrument was replaced with another identical instrument (SN 1450) in 2019, the parallel profiles were repeated on August 26th, 2020 and September 7th, 2020. All conductivity profiles obtained in these experiments are shown in Fig. A1a and b. It should be acknowledged that the latter of the SAIV instruments (SN 1450) showed relatively large offset for the conductivity sensor one year after being laboratory-calibrated before purchase in 2019. We note that also Ericson et al. (2018) detected that salinities from SAIV A/S SD204 were 0.10-0.15 PSU higher compared to the salinities measured by a Sea-Bird Electronics CTD device.

Based on the conductivity comparison presented in Fig. A1 a linear dependency was detected for the offset and therefore a first-degree polynomial $C_{corrected} = aC_{measured} + b$ was used to calculate the corrected conductivity. The data points used to determine the coefficients are shown in Fig. A1c and d. For the conductivity sensor of SAIV A/S SD208 SN 1269 coefficients $a = 0.9977$ and $b = -0.1051$ yielded corrections ranging from -0.16 mS cm$^{-1}$ to -0.18 mS cm$^{-1}$ for conductivities 24 mS cm$^{-1}$ and 32 mS cm$^{-1}$, respectively. Correction for the temperature sensor was calculated identically, $T_{corrected} = aT_{measured} + b$. With coefficients $a = 1.0343$ and $b = -0.0296$ the reduction was 0.03 degree to temperature of 0 °C and 0.04 °C to temperature of 2 °C.

The detected correction for the conductivity sensor of SAIV A/S SD208 SN 1450 used in the monitoring campaign from 2019 onward was somewhat larger compared to SN 1269 and displayed stronger linear dependency. The corrections were -0.21 mS cm$^{-1}$ to -0.36 mS cm$^{-1}$ for conductivities 24 mS cm$^{-1}$ and 32 mS cm$^{-1}$, respectively (coefficients for the polynomial being $a = 0.9805$ and $b = 0.2623$). A smaller linear correction with coefficients 0.9963 and 0.0199 was applied to the temperature sensor. This correction was around 0.02 °C for temperature range -2 to 2 degrees. For higher temperatures the correction was within the limits of the sensor accuracy (Table 2).

After the offset for the SAIV conductivity sensor was last determined in autumn 2020, consistency checks have been conducted whenever independent measurements from different instruments are available and suspicious data is removed.

*Author contributions.* As the principal investigator MM was responsible for initiating, designing and executing the monitoring program in Hornsund fjord. MM and OG have developed the monitoring program into its current form. All authors took part in the data collection. The post-processing of the CTD data and the curation of all data were done by MK and MM. MK prepared the original manuscript and figures. All authors contributed by reviewing and editing the manuscript.

*Competing interests.* The authors declare no competing interests.

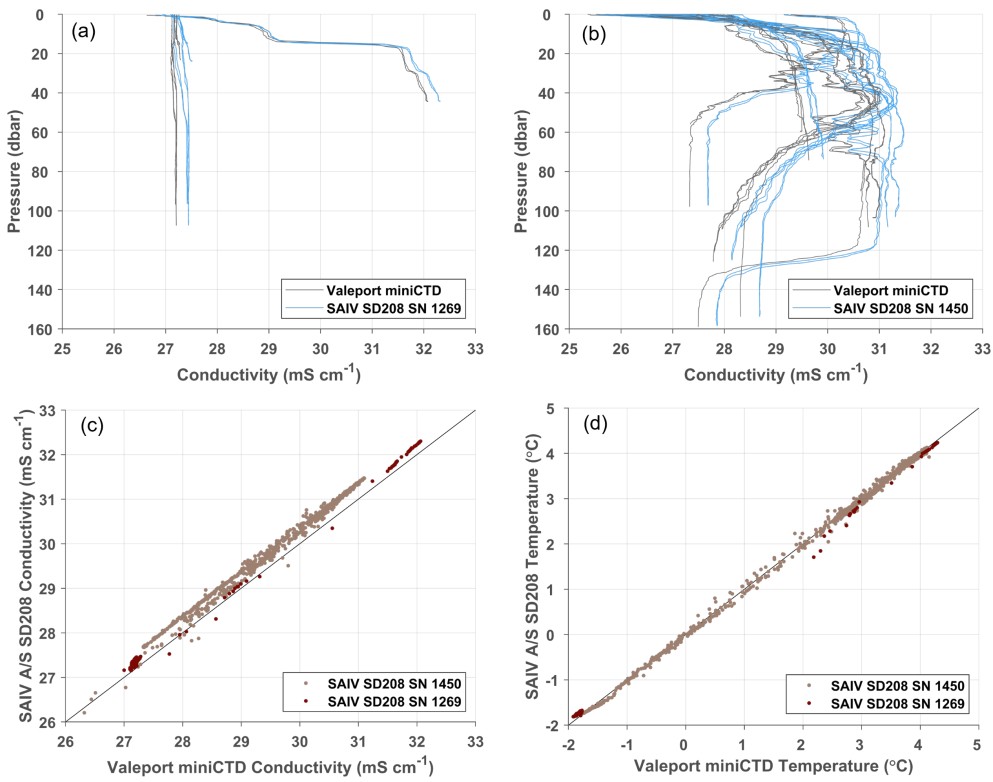

**Figure A1.** Results of the intercalibration of the SAIV A/S SD208 CTD sensors. Conductivity profiles taken on September 22nd, 2017 and April 13th, 2018 with SAIV SN 1269 (a) and taken on August 26th, 2020 and September 7th, 2020 with SN 1450 (b) are shown together with parallel profiles with Valeport mini CTD. Data points used for defining the coefficients of the linear first-degree polynomial used for the correction are shown for conductivity (c) and temperature (d).

*Acknowledgements.* The authors would like to thank the staff of the Polish Polar Station Hornsund and personnel of IG PAS helping with realization of the monitoring program, data collection and archival. In particular the authors are grateful of the time and effort of the designated oceanographers Maciej Błaszkowski, Mariusz Czarnul, Łukasz Pawłowski, Michał Niedbalski, Kacper Wojtysiak, Tomasz Lenz, and
Aleksander Sikorski who participated in the data collection.

The LONGHORN monitoring program is financed by financed by the Ministry of Science, Poland (grant/award nos. 3841/E-41/SPUB/2015, 3841/E-41/SPUB/2016/1, 15/E-41/SPUB/SP/2019, and 3/524698/SPUB/SP/2022). Data compilation and analysis as well as part of the monitoring during years 2022 and 2023 were funded by the Norwegian Financial Mechanism 2014-2021 (grant no. UMO-2019/34/H/ST10/00504), National Science Centre, Poland (grant no. UMO-2020/39/B/ST10/01504) and Argo-Poland funded by the Polish Ministry of Science
(grant/award no. 2022/WK/04).

MK, MM, OG and VJ were funded by Norwegian Financial Mechanism 2014-2021 (grant no. UMO-2019/34/H/ST10/00504). MK, MM and OG were funded by Argo-Poland (grant/award no. 2022/WK/04). OG and MM have been supported by the National Science Centre, Poland (grant no. 2021/43/D/ST10/00616).





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
