# Peer review of "Oceanographic monitoring in Hornsund fjord, Svalbard"

_Earth System Science Data, 2024_

## Author Response (AR1)

**RESPONSE TO RC1**

*The authors thank the reviewer for the thorough inspection of the manuscript. We are especially thankful for the time and effort used to check the data files and to point out the flaws found in them. Please find our answers to the specific comments below.*

**Specific comments**

Please follow TEOS-10 guidelines: "**published values of salinity should be specifically identified as being either Practical Salinity with the symbol S$_P$ or Absolute Salinity with the symbol S$_A$**" (citation from https://www.teos-10.org/). This should be done throughout the manuscript as well as in the data files.

*We have clarified the distinction in the manuscript. However, the PANGAEA database where the data files are stored, does not use Practical Salinity as a variable name, but refers to it as Salinity. Therefore changing the variable name in the dataset would require changing the database. Nonetheless, we consider the policy of the database, together with our choice of discussing only practical salinity and in situ temperature in the manuscript, to be in line with the TEOS-10 recommendations. In the TEOS-10 manual (https://www.teos-10.org/pubs/TEOS-10_Manual.pdf, page 10) it is stated that* **"Data stored in national and international data bases should, as a matter of principle, be measured values rather than derived quantities. Consistent with this, we recommend continuing to store the measured (in situ) temperature rather than the derived quantity, Conservative Temperature. Similarly we strongly recommend that Practical Salinity PS continue to be the salinity variable that is stored in such data bases since PS is closely related to the measured values of conductivity."**

Line 42: Here, you mention Winter Cooled Water for the first time. Please describe this water mass briefly for the reader not familiar with the area. *The Winter Cooled Water is now described.*

Line 136-139: In line 122 you say "over 50" stations, but here they are about 100. It is not clear how many belong to the monitoring program and how many to "various research projects". Please clarify. *The other 50 stations were related to projects ongoing in Hansbukta and this has been clarified in the text.*

Line 182-183: I find that you are very specific here, mentioning subjective commitment of persons. I would think that this falls within "availability of staff". Please delete or rewrite. *This has been deleted.*

Line 235: I find it weird that low salinity values mostly occur, when there is no freshwater flux to the fjord. Please clarify. *This is assumed to be due to ice formation in the conductive cell and it is now clarified in the text.*

Line 243: On what basis did you choose these lower limits? Please clarify. *The limits were chosen by visually inspecting all data appearing to lay outside of the general spread. This is now clarified in the text.*

**Technical corrections**

Check the use of comma's. E.g. line 41 "In recent years, warming …"; line 122 "Initially, the hydrographic …" and elsewhere. *We have inspected the text and added commas in the places indicated and in a few other places.*

Line 37: "… relatively little studied." Check language and consider rephrasing. *The sentence has been rephrased.*

Line 47: "… to occupy larger extent…" Check language and consider rephrasing. *The sentence has been rephrased.*

Line 70 and 74: There are more than one survey, right? Thus, should be "Hydrographic surveys…. have" and "The surveys contribute to…" *This has been changed.*

Line 123 and 127: Table 1 only includes 11 stations. I suggest to move the reference to Table 1 from line 123 to line 127. *The reference has been moved as suggested.*

Line 194: replace "early years" with "first years". *This has been replaced.*

Line 279: delete "one" in "the other one" *This has been deleted.*

Line 312: Insert space in mgl$^{-1}$. *The space has been inserted.*

Line 436: Delete one of the "financed by". *This has been deleted.*

Figure 1, legend: You describe the yellow star in the text for 1c, but the star is much more visible in 1b. Please move this part to text for 1b. *This has been done.*

Table 1. second line: "between" instead of "netween". Comma after "In addition, " *These corrections have been made.*

Figure 2: Here, you are plotting data from stations BuBW_01 and BuBW_04, but I cannot find station BuBW_04 in Figure 1. Please add it to the map. *The station BuBW1_04 has been marked on the map. In addition all locations of the core stations before 2022 are indicated (when the distance is large enough for them to be visible under the new/current locations).*

Figure 4: The plus-signs in (b) are not very visible and it is hard to see their colours. Please change to a filled symbol. *Filled symbols are now used in Fig. 4b.*

There is no reference to Appendix A. Please indicate the offset issue in the text and add reference to the appendix. *The issue is now referred to in Section 3.1.1 with reference to the appendix.*

**Specific comments on data**

Hornsund-fjord_CTD: When plotting all oxygen data with depth, one profile stands out with increasing oxygen with depth. This is visible in the depth range ~75m to ~170m, where the profile ends with two lower values. Please check the data.

*We agree that this profile looks suspicious and it has been removed from the dataset.*

Hornsund-fjord_CTD_suppl: The variable "Depth Water" seems to be identical to "Temp". This is a serious error.

The Dissolved oxygen concentration is missing.

*Actually the data for "Depth Water" was missing, causing the following columns to shift to the left. The data has now been inserted, which fixed both of the above mentioned errors.*

Hornsund-fjord_SR: A minor issue: I was not able to import the data to Matlab using the import tool, as only the first two columns were visible. After deleting the header info, the problem was solved. Please check, if there is an issue in the shift from header to data. The other files were okay.

*We have reported this issue to the database administrator responsible for creating the final datafiles.*

*The authors are grateful for the reviewer for the insightful comments and the suggestions for improving both the manuscript as well as the monitoring program and the quality of measurements. As stated in the manuscript, we were not aware of the offset in the SAIV conductivity sensor before 2019 and even when the offset was found out, we assumed it applied to one instrument and/or one faulty calibration and that the same problem would not apply to the new instrument. However, in 2023 the monitoring program used RBR CTDs in parallel with SAIV and since 2024 the main instrument for the monitoring is RBR CTD. In addition to turbidity and oxygen, the RBR instruments include sensors for measuring $pCO_2$, pH, PAR and Chlorophyll A in order to include biological and chemical observations in the monitoring program. Because of the additional parameters and only one year of data, the data recorded by the RBR instruments are not discussed in this manuscript.*

*We have now tried to address the reviewer's wishes to briefly discuss the large-scale importance and effects of hydrographic changes in the introduction and conclusion parts of the manuscript. In addition, a sentence and couple references on the ecosystem/biology in Hornsund/West Spitsbergen fjords has been added.*

*The reviewer is correct that with the sediment measurements the aim is to catch the runoff and the terrestrial matter discharge from land to ocean. With the amount of organic matter, the data also shows variability in biological production. We have now replaced the first sentence of the section 3.2 to the introduction and added a sentence in order to be clearer with our motivation to monitor sedimentation and suspended sediment concentration.*

*Unfortunately, we cannot use the Rossby radius as a quantitative argument for the chosen spacing of the CTD stations. In relatively shallow coastal waters around Svalbard the Rossby radius varies with seasons from 1 to 7 km. The distances between the current monitoring stations are larger than that, suggesting that it would be necessary to have a more densely spaced array than is currently in use. However, besides of the visual inspection of the collected data, the amount of monitoring stations was also based on the time and effort required to maintain the desired frequency of measurements. We have now added a sentence clarifying this.*

*Our responses to the specific comments can be found below.*

172-173: The BUBW01 example used to state that the horizontal differences are small seems to be from a really small side arm. Is this a representative example?

*We have chosen BuBW_01 as an example because the distance between the old (pre 2022) and new (since 2022) locations is the largest (1.5 km, Table 1) for this station, and therefore it is plausible to assume the horizontal differences in water properties are also the largest for this station. In Fig. 2 we compare stations in the Western Burgerbukta with distance of 2.2 km. Most of the stations measured since 2022 are from the side arms and only one station, FC_01, is from the central fjord. However, the location of the FC_01 changed only by 500 m in 2022. We have clarified this choice in the text now.*

182 "…subjective commitment of the person…" sounds quite unprofessional. Is the irregular sampling coverage going to affect upcoming analyses?

*This has been removed. The irregular sampling coverage is possibly a challenge for some analyses and therefore the aim of the monitoring program is to cover the whole period from May to October each year. Unfortunately, due to circumstances such as COVID-19 pandemic, this has not been possible each year.*

205 remind the reader of the water depths at the 2 sediment trap sites here

*The depths have now been included in the first paragraph of the section where the stations for water sampling and sediment traps are introduced.*

229 it probably would be more up to date to present the data as conservative temperature and absolute salinity

*We are aware of this, but as, for example, the water mass definitions in the West Spitsbergen fjords are based on practical salinity and in situ/ potential temperature (e.g. Nilsen et al., 2008: Promińska et al., 2018), we have decided to use them. Continuing to use practical salinity and in situ temperature as the main parameters in the stored datasets is also a recommendation of TEOS-10. In the TEOS-10 manual (https://www.teos-10.org/pubs/TEOS-10_Manual.pdf, page 10) it is stated that "**Data stored in national and international data bases should, as a matter of principle, be measured values rather than derived quantities. Consistent with this, we recommend continuing to store the measured (in situ) temperature rather than the derived quantity, Conservative Temperature. Similarly we strongly recommend that Practical Salinity PS continue to be the salinity variable that is stored in such data bases since PS is closely related to the measured values of conductivity**."*

248-252 about running the sensor into the bottom: perhaps think of using a 5m (or so) lead line with a weight below the CTD. This way the user can feel once the weight is at the seafloor and reduce the lowering speed to avoid running the CTD into the ground, which apparently affects the data quality

*This is a good suggestion and currently we are using such a system with the RBR instruments, which have the sensors in the bottom. As described in the text, we did not consider this necessary with the SAIV CTD which had the sensors in the top part of the instrument, but it would have been good practice with the Valeport miniCTD. However, as the Valeport miniCTD had a high enough sampling frequency, we did not need to include the up-profile, which was compromised by the bottom contact with the sediments, for the final data.*

253 "the data are compressed to 1 dbar". What does that mean exactly? Sampling with 0.5 Hz and a lowering speed of 0.4m/s (but often faster) results in 1 or less sample per meter. This seems unnecessarily coarse and might be one argument to use other CTDs.

*We are aware of the coarseness of the raw data and therefore we have included also the up-profile for the averaged data, which typically increases the number of measurements to 2 or 3 per meter. This has now been clarified in the text.*

Fig 1: PPS and the yellow star is not easy to find

*The star is now made bigger.*

Fig 6, 285-294: I find that the characteristics described in the text are not entirely easy to see in the figures.

299-300: also really difficult to see. Might be the choice of colors or maybe it's just my eyes…

*As the aim is here to present merely an overview of the data, not analysis, we chose not to change the figure in order to better illustrate the discussed details. Instead we rewrote the text to focus more on the general ranges.*

Fig. 7: why is Hansbukta chosen? Are tides important there? In such a shallow bay with tides and other mixing mechanisms it may not be surprising to have a homogeny water body. The vertical lines are kind of difficult to see, at least in my version.

*Hansbukta was chosen as an example simply because the seasonal coverage is in general the longest there due to its closeness to the Polish Polar Station. This has been clarified in the text and also the importance of tidal mixing is now mentioned.*

Fig6/7: is there really no sea ice at all in Hornsund during winter? The near-freezing temperatures do seem to suggest complete overturning and ice formation during winter and stratification and warming during summer.

*We are not using these figures to make statements on the absence of sea ice, but on the contrary, agree that the water properties show ice formation during winter and established stratification during summer. Figure 6 shows data only from August, but also in this figure the influence of ice formation is seen as the remnants of cold Winter Cooled Water close to the bottom. We have now clarified this in the text.*

345 If knowing the exact depth is important, one could attach a small pressure sensor to the device

*Occasionally the CTD has been used for this purpose when taking water samples. However, knowing the exact depth is not as important as sampling from constant depths. Here we wanted merely to acknowledge possible sources for uncertainties.*

399 Seabird not Seebird

*This has been corrected.*